# Hemopexin and $\alpha_1$-microglobulin heme scavengers with differential involvement in preeclampsia and fetal growth restriction

Lina Youssef[1,2]*, Lena Erlandsson[1], Bo Åkerström[3], Jezid Miranda[2], Cristina Paules[2], Francesca Crovetto[2,4], Fatima Crispi[2,4], Eduard Gratacos[2,4], Stefan R. Hansson[1,5]

**1** Section of Obstetrics and Gynecology, Department of Clinical Sciences Lund, Lund University, Lund, Sweden, **2** BCNatal | Fetal Medicine Research Center (Hospital Clínic and Hospital Sant Joan de Déu), Institut d'Investigacions Biomèdiques August Pi i Sunyer (IDIBAPS), University of Barcelona, Barcelona, Spain, **3** Section of infection Medicine, Department of Clinical Sciences, Lund University, Lund, Sweden, **4** Centre for Biomedical Research on Rare Diseases (CIBER-ER), Madrid, Spain, **5** Skåne University Hospital, Lund/Malmö, Sweden

* lyoussef@clinic.cat, linayoussefdr@gmail.com

**Data Availability Statement:** All relevant data are within the manuscript and its Supporting Information files.

## Abstract

Hemopexin and $\alpha_1$-microglobulin act as scavengers to eliminate free heme-groups responsible for hemoglobin-induced oxidative stress. The present study evaluated maternal and fetal plasma concentrations of these scavengers in the different phenotypes of placenta-mediated disorders. Singleton pregnancies with normotensive fetal growth restriction [FGR] (n = 47), preeclampsia without FGR (n = 45) and preeclampsia with FGR (n = 51) were included prospectively as well as uncomplicated pregnancies (n = 49). Samples were collected at delivery and ELISA analysis was applied to measure the hemopexin and $\alpha_1$-microglobulin concentrations. In maternal blood in preeclampsia with and without FGR, hemopexin was significantly lower (p = 0.003 and p<0.001, respectively) and $\alpha_1$-microglobulin was significantly higher (p<0.001 in both) whereas no difference existed in normotensive FGR mothers compared to controls. In contrast, in fetal blood in growth restricted fetuses with and without preeclampsia, both hemopexin and $\alpha_1$-microglobulin were significantly lower (p<0.001 and p = 0.001 for hemopexin, p = 0.016 and p = 0.013 for $\alpha_1$-microglobulin, respectively) with no difference in fetuses from preeclampsia without FGR in comparison to controls. Thus, hemopexin and $\alpha_1$-microglobulin present significantly altered concentrations in maternal blood in the maternal disease -preeclampsia- and in cord blood in the fetal disease -FGR-, which supports their differential role in placenta-mediated disorders in accordance with the clinical presentation of these disorders.

## Introduction

Preeclampsia (PE) and fetal growth restriction (FGR) constitute a great challenge in modern obstetrics since they affect 2–5% and 5–10% of pregnancies, respectively [1, 2]. PE is responsible for significant maternal and perinatal morbidity and mortality worldwide [3]. In addition,

**Funding:** This project has been partially funded with support of the Erasmus + Programme of the European Union (Framework Agreement number: 2013-0040). This publication reflects the views only of the authors, and the Commission cannot be held responsible for any use, which may be made of the information contained therein. Additionally, the research leading to these results has received funding form "la Caixa" Foundation under grant agreements LCF/PR/GN14/10270005 and LCF/PR/GN18/10310003, the Instituto de Salud Carlos III (PI14/00226, PI15/00130, PIE15/00027, PI17/00675, PI18/00073) integrados en el Plan Nacional de I+D+I y cofinanciados por el ISCIII-Subdirección General de Evaluación y el Fondo Europeo de Desarrollo Regional (FEDER) "Una manera de hacer Europa", Cerebra Foundation for the Brain Injured Child (Carmarthen, Wales, UK) and AGAUR 2017 SGR grant n° 1531. This work was also funded by the Swedish Medical Research Council (VR), governmental ALF research grant to Lund University and Lund University hospital, the Royal Physiographic Society in Lund and the Swedish Foundation for Strategic Research (SSF).

**Competing interests:** SH and BÅ hold a patent related to the subject in this paper (PCT appl no. WO2008098734A1; Diagnosis and treatment of preeclampsia). SH and BÅ are co-founders and share-holders of Guard Therapeutics formerly named A1M Pharma. This company develops a treatment of acute kidney injury based on $\alpha_1$-microglobulin, but is not involved in clinical development of diagnosis and treatment of preeclampsia, and has not supported this study financially. This does not alter our adherence to PLOS ONE policies on sharing data and materials. The remaining authors declare that the research was conducted in the absence of any commercial or financial relationships that could be construed as a potential conflict of interest.

FGR has a great burden on fetal well-being and half of the stillbirths are associated with FGR [4]. The exact pathogenesis of PE and FGR is still poorly understood which prevents the development of specific treatments that may prolong the pregnancy and minimize the complications of these disorders. Both of PE and FGR are considered being placenta-mediated disorders since they are characterized by insufficient perfusion of the placenta associated with oxidative stress and local hypoxia [5–7]. Recently, cell-free hemoglobin including free fetal hemoglobin (HbF) has been reported to be related to PE and concomitant FGR [8, 9]. An up-regulation of HbF gene expression has been observed in placentas from PE pregnancies together with placental HbF accumulation most likely secondary to hypoxia [10]. Additionally, the experimental evidence of human placental hemoglobin perfusion resulted in physiological and morphological changes similar to PE placentas [10, 11]. Cell-free hemoglobin has pro-oxidant and inflammatory properties [12] and is associated with endothelial dysfunction and cardiovascular complications in hemolytic disorders [13]. Increased cell-free hemoglobin levels have also been reported in maternal plasma from pregnancies complicated by PE [7, 14] and cord blood from fetuses affected by FGR associated with PE, overall hemoglobin-induced oxidative stress was proposed to contribute to the development of these conditions [9, 15].

To prevent the toxic effect of free hemoglobin and its degradation compounds heme and free iron, the human body disposes of several circulating scavenger proteins. Haptoglobin is the main well studied hemoglobin scavenger, that collects most of the cell-free hemoglobin and carries it to macrophages and hepatocytes [16]. Another important heme scavenger is hemopexin, which is an acute phase glycoprotein synthesized by the liver that has a high affinity to bind free heme [17]. When the haptoglobin system gets saturated, hemopexin frees the blood from heme [18]. In addition, $\alpha_1$-microglobulin (A1M) is an alternative heme scavenging protein that binds heme and radicals and has enzymatic neutralizing capability. Recently, A1M has been shown to be up-regulated in early [19, 20] and late-onset PE [14, 21]. However, there is still a knowledge gap regarding the involvement of the heme scavengers in PE with and without FGR as well as in normotensive FGR.

Thus, the objective of this study was to evaluate paired maternal and fetal plasma samples for concentrations of hemopexin and A1M in the different phenotypes of PE and/or FGR compared to uncomplicated pregnancies.

## Methods

### Study population

This was a prospective observational study including singleton pregnancies with a diagnosis of PE and/or FGR that attended the Departments of Maternal-Fetal Medicine at BCNatal (Barcelona, Spain) between January 2016 and December 2017. Fetal growth restriction was defined as estimated fetal weight (EFW) and birthweight below the $10^{th}$ centile associated with either abnormal cerebroplacental ratio ($<5^{th}$ centile) or abnormal uterine arteries mean pulsatility index ($>95^{th}$ centile), or birthweight below the $3^{rd}$ centile [22]. The EFW and birthweight centiles were assigned according to local standards [23]. Preeclampsia was defined as high blood pressure (systolic blood pressure $\geq$140 mmHg and/or diastolic blood pressure $\geq$90 mmHg on two occasions, at least four hours apart) that developed after 20 weeks of gestation, and was combined with proteinuria ($\geq$300 mg/24 hours or protein/creatinine ratio $\geq$0.3) [1, 24]. Uncomplicated pregnancies with normotensive mothers and appropriate growth for gestational age fetuses -defined as EFW and birthweight above the $10^{th}$ centile- were randomly selected from our general population to be included as controls and frequency paired with cases. Spontaneous preterm deliveries without clinical signs of infection, or iatrogenic preterm deliveries due to placenta previa were recruited as preterm controls. Preterm deliveries were

considered when gestational age at delivery was $\geq$24 and <37 weeks gestation. Term deliveries included pregnancies with gestational age at delivery $\geq$37 and <42 weeks of gestation. In all pregnancies, gestational age was calculated based on the crown-rump length at first trimester ultrasound [25]. Pregnancies with chromosomal/structural anomalies or intrauterine infection were excluded. The study was conducted in accordance with the principles of the Helsinki declaration and the relevant guidelines and regulations. The study protocol was approved by the ethical committee of clinical research at Hospital Clínic, Barcelona, Spain (HCB/2016/0253) and patients accepting to participate provided their written informed consent.

## Data collection and study protocol

The following data were recorded upon enrollment: maternal age, ethnicity, body mass index (BMI), known chronic disease (i.e., hypertension, diabetes mellitus), parity, obstetric history, mode of conception and smoking status. The feto-placental Doppler parameters were obtained in the last 2 weeks of pregnancy. Ultrasound studies were performed using a Siemens Sonoline Antares (Siemens Medical Systems, Malvern, PA, USA) or a Voluson 730 Expert (GE Medical Systems, Milwaukee, WI, USA) with 6–4-MHz linear curved-array probes. The EFW was calculated using the Hadlock formula [26] and centile was based on local reference curves [23]. All estimations were done in the absence of fetal movements and, when required, with the mother in voluntary suspended respiration. An angle of insonation of <30˚ between the vessel and the Doppler beam was accepted for analysis. The mechanical and thermal indices were maintained below 1, and the wall filter was set to 70 Hz. Feto-placental Doppler parameters were obtained from three or more successive waveforms in each vessel. Doppler examination included uterine arteries (UtA) [27], umbilical artery (UA) [28] and the fetal middle cerebral artery (MCA) pulsatility indices (PI) [28], with the calculation of the cerebroplacental ratio (CPR) [29]. These values were normalized into z-scores accordingly [27–29].

At time of delivery, maternal serum creatinine, admissions to the intensive care unit, gestational age, birthweight, birthweight centile, Apgar scores, umbilical artery pH, admissions to the neonatal intensive care unit and perinatal mortality were recorded. In addition, paired maternal and cord blood were collected to measure the concentrations of Hemopexin and A1M. Pregnancies with missing perinatal outcomes or unconfirmed FGR diagnosis (a small newborn not fulfilling FGR definition with birthweight between the 3rd and the 10th centile associated with normal cerebroplacental ratio ($\geq$5th centile) and normal uterine arteries PI ($\leq$95th centile)) [22] were excluded as shown in Fig 1.

## Follow up and management

All pregnancies with FGR diagnosis were monitored fortnightly with fetal growth evaluation, amniotic fluid assessment and Doppler measurements. The management of these pregnancies relied on a standardized protocol [22]. Indications for labor induction were as follows: 1) at $\geq$26 weeks' gestation: non-reassuring cardiotocography register and/or reversed ductus venosus (DV) diastolic flow 2) at $\geq$30 weeks' gestation, one or more of the following: UA reversed end diastolic volume, DV-PI $\geq$95th centile, DV absent diastolic flow 3) at $\geq$34 weeks' gestation: UA absent end diastolic volume 4) at $\geq$37 weeks' gestation, one or more of the following: estimated fetal weight <3rd centile, persistent (12 h apart) MCA-PI <5th centile or UA-PI > 95th centile or CPR <5th centile, UtA-PI >95th centile 5) at $\geq$40 weeks' gestation, estimated fetal weight $\geq$3rd and <10th centile with normal Doppler parameters. In the first three scenarios, termination of pregnancy was by cesarean section, whereas in the next two by labor induction.

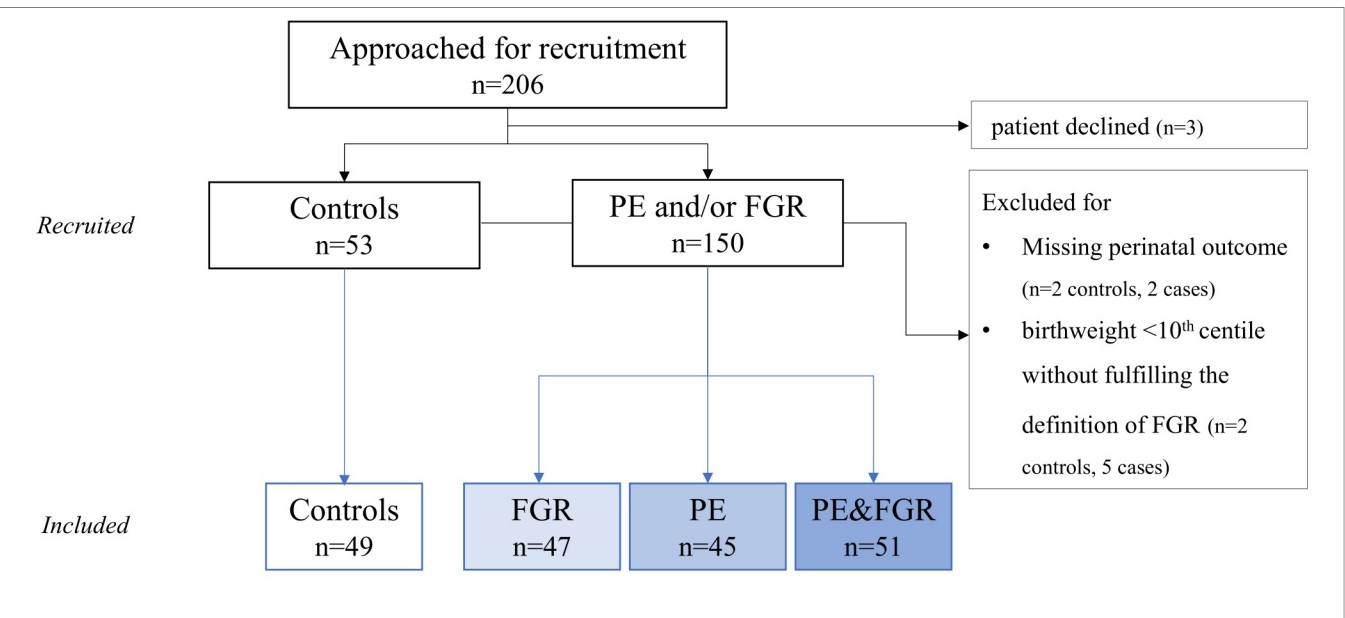

**Fig 1. The flow chart of the study.** FGR, fetal growth restriction; PE, preeclampsia.

Pregnancies diagnosed with PE were classified as mild (to be terminated at $\geq$37 weeks) and severe (to be terminated at $\geq$34 weeks or before if applies). Severe PE was considered according to the criteria of the American College of Obstetricians and Gynecologists [24]. Pregnancy termination was by labor induction or cesarean section upon obstetric indication.

Labor induction was achieved by hormonal cervical ripening with a slow-release prostaglandin E2 vaginal pessary (10mg) or mechanical with foley catheter. Oxytocin induction was indicated thereafter for failure of labor onset within 18h. During the labor course, cesarean or operative vaginal delivery was indicated for non-reassuring fetal status, based on abnormal fetal heart rate tracing [30] and adverse fetal scalp blood pH during intrapartum monitoring.

## Paired maternal and cord blood sampling

Maternal blood samples were drawn from peripheral veins around delivery (up to a maximum of 2 hours after delivery). Cord blood was obtained from the umbilical arteries after the cord was clamped at delivery. All blood samples were collected in EDTA-treated tubes. Plasma was separated by centrifugation at 1500 g for 10 min at 4°C, and samples were immediately stored at −80°C until analyzed.

## Heme scavengers and other biomarkers

**Hemopexin concentrations.** Hemopexin concentrations were determined using a Human Hemopexin ELISA Kit GWB-4B6D1A from Genway Biotech Inc (San Diego, CA, USA) according to manufacturer's instructions. Standards and unknown samples were run in duplicates and the absorbance was read at 450nm using a Wallac 1420 Multilabel Counter (Perkin Elmer Life Sciences, Waltham, MA, USA).

**A1M concentrations.** An in-house developed sandwich ELISA was used for the quantification of A1M. Microtiter plates were coated with an anti-human A1M antibody (mouse monoclonal, clone 35.14; 5 µg/ml in PBS) overnight at +4°C under sealing film with 100 ml/well. After washing three times with PBS C 0.05% tween-20, 100 ml of human urinary A1M

reference standard samples (1.56–100 ng/ml in PBS C 0.05% tween-20) or unknown plasma samples (diluted 1000x with PBS C 0.05% tween-20) were added to the wells and incubated under sealing film for 1 h at room temperature, darkness and rotational shaking 250–500 rpm. After washing three times with PBS C 0.05% tween-20, 100 ml/well of a horseradish peroxidase (HRP)-conjugated detection antibody (mouse monoclonal, clone 57.10; 5 ng/ml) was added and incubated under sealing film for 1 hour at room temperature, darkness and rotational shaking 250–500 rpm. Finally, after washing three times with PBS C 0.05% tween-20, 100 ml/well of a ready-to-use 3,3´,5,5´-Tetramethylbenzidine (TMB, Life Technologies, Stockholm, Sweden) substrate solution was added, sealed, and incubated without shaking. The reaction was stopped after 20 minutes using 1 M sulfuric acid and the absorbance was read at 450nm using a Wallac 1420 Multilabel Counter (Perkin Elmer Life Sciences, Waltham, MA, USA). The samples were analyzed in 10 batches (5 for each of maternal and fetal plasma samples). A standard curve was used to quantify the A1M concentration in each sample with the preparation of a new standard curve for each plate (an example is presented as S1 Fig in S1 File). The average intraassay coefficient of variation was 2.23% (±1.8% standard deviation) and 1.21% (±0.95% standard deviation) for maternal and fetal A1M respectively in harmony with previous studies [20, 31]. Both anti-human A1M and HRP-conjugated detection antibodies produced in-house by Agrisera AB (Vännäs, Sweden) by immunization with human urinary A1M prepared as previously described [32].

## Additional biochemical measurements in cord blood samples

To investigate the presence of cell-free hemoglobin in cord blood samples, uncomplexed free HbF levels were quantified by an in-house developed sandwich ELISA. Microtiter plates were coated with anti-human HbF antibody (mouse monoclonal, clone 13-47-1; 20µg/ml) overnight at +4˚C. Standards and unknown samples were added in duplicates and incubated for 2 hours at room temperature. A biotinylated detection antibody (monoclonal antibody, clone102.11; 4µg/ml) was added and incubated for 1 hour at room temperature. Streptavidin-HRP working solution cat no S5512; 0.5mg from Sigma-Aldrich was added and incubated for 30 minutes at room temperature. Finally, a ready-to-use TMB (Life Technologies, Stockholm, Sweden) substrate solution was added. The reaction was stopped after 20 minutes using 1 M sulfuric acid solution and the absorbance was read at 450nm using a Wallac 1420 Multilabel Counter (Perkin Elmer Life Sciences, Waltham, MA, USA). A standard curve was used to quantify the HbF concentration in each sample. Both anti-human HbF antibody and the biotinylated detection antibody were prepared in house by immunization with purified human hemoglobin gamma-chains [14]. Lactate dehydrogenase (LDH) activity was measured to validate the level of mechanical hemolysis due to sampling using an LDH Assay kit ab102526 from Abcam (Cambridge, United Kingdom). Erythropoietin was measured to evaluate any expressional changes linked to changes in plasma free HbF levels that might have occurred *in vivo*. A fully automated chemiluminescent immunoassay for erythropoietin on an Immulite 2000 analyzer (Siemens Healthineers) was used following the manufacturer's instructions.

## Statistical analysis

Data were analyzed with the statistical software STATA 14.2 (StataCorp LLC, Texas, USA). A sample size of 35 patients per group was calculated by expecting differences of 3 standard deviations in hemopexin concentrations between each group of the cases and the controls (both in maternal and cord blood) [9, 21], for a given 5% α error and 80% power.

Results were expressed as mean (± standard deviation), median (interquartile range) or percentage, as appropriate. Statistical analysis included the use of student *t* or Mann Whitney U

tests for continuous variables and Pearson $\chi^2$ test for the categorical ones. Each group of the cases was compared separately with controls. To evaluate the influence of confounding factors, comparisons were adjusted for the differences in ethnicity, chronic hypertension, diabetes, smoking, fetal sex and gestational age at time of sampling by multiple regression analyses. In addition, a sub analysis was performed by dividing the population according to gestational age at delivery to preterm or term and also to pregnancies with a male or female fetus. To explore the correlation between maternal hemopexin or A1M and maternal serum creatinine on one hand, and fetal hemopexin or A1M on the other hand, Pearson correlation coefficient was calculated. Maternal and fetal hemopexin and A1M correlation with pregnancy outcome has been also explored including time to delivery, case severity, admissions to the intensive care unit, APGAR score at 5 minutes and neonatal intensive care unit admissions. All reported p values are two-sided. Differences were considered significant when $p<0.05$.

## Results

Table 1 displays the characteristics of the study population which consisted of 192 patients. The study groups were comparable regarding maternal age, parity and mode of conception. Chronic hypertension and pregestational BMI were significantly higher among PE cases, whereas a higher prevalence of smokers was observed in normotensive FGR with lower pregestational BMI, in comparison to controls. As expected, higher uterine and umbilical arteries PI were present in FGR groups with lower fetal MCA-PI and CPR. In addition, gestational age at time of delivery was earlier in PE and/or FGR.

The heme scavenger results are shown in Fig 2 and in S1 Table in S1 File. In maternal blood, hemopexin was significantly lower in PE with and without FGR ($p = 0.03$, $p<0.001$) respectively when compared to controls. In addition, significantly higher concentrations of A1M ($p<0.001$) were seen in both groups of PE than controls. Similar concentrations of hemopexin or A1M concentrations were seen in normotensive FGR and controls ($p = 0.48$ for hemopexin, $p = 0.54$ for A1M). In fetal blood, hemopexin and A1M were significantly lower in FGR fetuses with and without PE compared to controls ($p<0.001$, $p = 0.001$ for hemopexin; $p = 0.016$, $p = 0.013$ for A1M) respectively, and with no difference seen for PE without FGR ($p = 0.06$ for hemopexin, $p = 0.85$ for A1M). These results remained significant after statistical adjustment for potential confounders such as ethnicity, chronic hypertension, pregestational diabetes, smoking, fetal sex and gestational age at sampling. In PE mothers, A1M was positively correlated with creatinine concentrations ($r = 0.33$, $p = 0.002$) in contrast to no correlation between hemopexin and creatinine levels ($r = -0.01$, $p = 0.92$) as displayed in S2 Fig in S1 File. The FGR fetuses had higher levels of free HbF, LDH activity and erythropoietin in their blood, compared to controls (S3 Fig in S1 File).

The subanalysis accounting for the differences in the gestational age at delivery showed comparable concentrations of maternal and fetal hemopexin and A1M in association with PE and/or FGR in preterm and term pregnancies (S2 and S3 Tables in S1 File). Additionally, male and female fetuses presented the same profile of heme scavengers in the different phenotypes of PE and/or FGR compared to controls (S4 and S5 Tables in S1 File).

No correlation existed between maternal and fetal hemopexin ($r = 0.12$, $p = 0.12$) or maternal and fetal A1M ($r = -0.01$, $p = 0.86$), when looking at the whole population or within the study groups.

None of maternal or fetal heme scavengers were correlated with time to delivery, maternal morbidity, including severity and days of admission at the intensive care unit, or perinatal morbidity such as low APGAR score at 5 minutes and neonatal intensive care admission.

**Table 1. Maternal, fetoplacental and perinatal characteristics of the study population.**

| | Controls | FGR | PE | PE&FGR |
|---|---|---|---|---|
| | N = 49 | N = 47 | N = 45 | N = 51 |
| *Maternal characteristics* | | | | |
| Age (years) | 35.1 (31.1–37.3) | 34.7 (29.3–38.8) | 33.6 (30.5–36.9) | 34.9 (31.6–37.6) |
| Caucasian ethnicity | 28 (57.1) | 36 (76.6)* | 20 (44.4) | 25 (49) |
| Pre-gestational BMI (Kg/m$^2$) | 22.9 (20.7–25.2) | 21.5 (19.8–23.3)* | 25.7 (22.4–29.3)* | 24.1 (21–26.4) |
| Chronic hypertension | 0 (0) | 0 (0) | 6 (13.3)* | 6 (11.8)* |
| Pre-gestational diabetes | 0 (0) | 3 (6.4) | 6 (13.3)* | 1 (2) |
| Nulliparity | 33 (67.4) | 27 (57.5) | 29 (64.4) | 36 (70.6) |
| Assisted reproductive technologies | 5 (10.2) | 6 (12.8) | 4 (8.9) | 5 (9.8) |
| Smoking during pregnancy | 4 (8.2) | 12 (25.5)* | 3 (6.7) | 5 (9.8) |
| *Feto-placental Doppler* | | | | |
| Uterine arteries mean PI (z score) | -0.22 (-1.26–0.94) | 0.87 (-0.21–2.76)* | -0.32 (-1.56–1.2) | 2.69 (1.96–3.45)* |
| Umbilical artery PI (z score) | -0.18 (-0.46–0.13) | 0.22 (-0.25–1.4)* | -0.31 (-0.53–0.12) | 0.6 (0.14–1.49)* |
| Middle cerebral artery PI (z score) | 0.01 (-0.32–0.92) | -0.36 (-1.05–0.13)* | -0.16 (-0.8–0.33) | -0.84 (-1.66––0.25)* |
| Cerebroplacental ratio (z score) | 0.01 (-0.56–0.86) | -0.95 (-2.32–0.06)* | -0.2 (-0.96–0.39) | -1.53 (-2.43––0.66)* |
| *Perinatal outcomes* | | | | |
| Gestational age at delivery (weeks) | 39 (36.4–40.3) | 37.6 (36.1–38.4)* | 37.3 (36.9–38.3)* | 35.1 (32.9–37.1)* |
| Deliveries <37 weeks gestation | 18 (36.7) | 13 (27.7) | 13 (28.9) | 34 (66.7) |
| Deliveries <34 weeks gestation | 3 (0.06) | 7 (14.9) | 2 (0.04) | 16 (31.4) |
| Cesarean section | 11 (22.5) | 23 (48.9)* | 29 (64.4)* | 29 (56.9)* |
| Male sex | 26 (53.1) | 25 (53.2) | 17 (37.8) | 31 (60.8) |
| Birthweight (g) | 3220 (2606–3664) | 2170 (1760–2438)* | 2948 (2660–3270) | 1794 (1474–2330)* |
| Birth weight centile | 56 (39–75) | 1 (0–2)* | 51 (21–81) | 1 (0–3)* |
| APGAR score 5 min <7 | 0 (0) | 0 (0) | 1 (2.2) | 2 (3.9) |
| Umbilical artery pH | 7.19 (7.15–7.24) | 7.23 (7.17–7.29) | 7.21 (7.17–7.24) | 7.23 (7.11–7.26) |
| Admission to neonatal intensive care unit | 9 (18.4) | 15 (31.9) | 12 (26.7) | 31 (60.8)* |
| Perinatal mortality | 0 (0) | 1 (2.2) | 0 (0) | 2 (3.9) |

Data are presented as median (interquartile range) or n (%) as appropriate. BMI, body mass index; FGR, fetal growth restriction; PE, preeclampsia. Perinatal mortality was defined as stillbirth or neonatal mortality within 28 days of delivery. The cerebroplacental ratio was calculated as the fetal middle cerebral artery PI divided by the umbilical artery PI.

* $p < 0.05$ by Mann Whitney U or Pearson $\chi^2$ tests as appropriate, compared to controls.

## Discussion

### Principal findings

This study demonstrates for the first time the differential involvement of heme scavengers hemopexin and A1M in the different phenotypes of PE and/or FGR by investigating their concentrations in paired maternal and fetal blood samples. Our results show that the heme scavenger systems have an altered profile in maternal blood in PE -with and without FGR- and in fetal blood in FGR -with and without PE- in comparison to uncomplicated pregnancies.

In the present study, we reveal that PE mothers have lower hemopexin along with higher A1M concentrations compared to controls. Conversely, fetuses who suffered FGR presented lower levels of hemopexin and A1M than fetuses from uncomplicated pregnancies. These results suggest that heme scavenger system is overwhelmed in PE mothers and FGR fetuses, possibly due to hemoglobin-induced oxidative stress [11, 14]. Interestingly, A1M behaved differently in maternal and fetal circulations, being elevated in the mother while it was reduced in

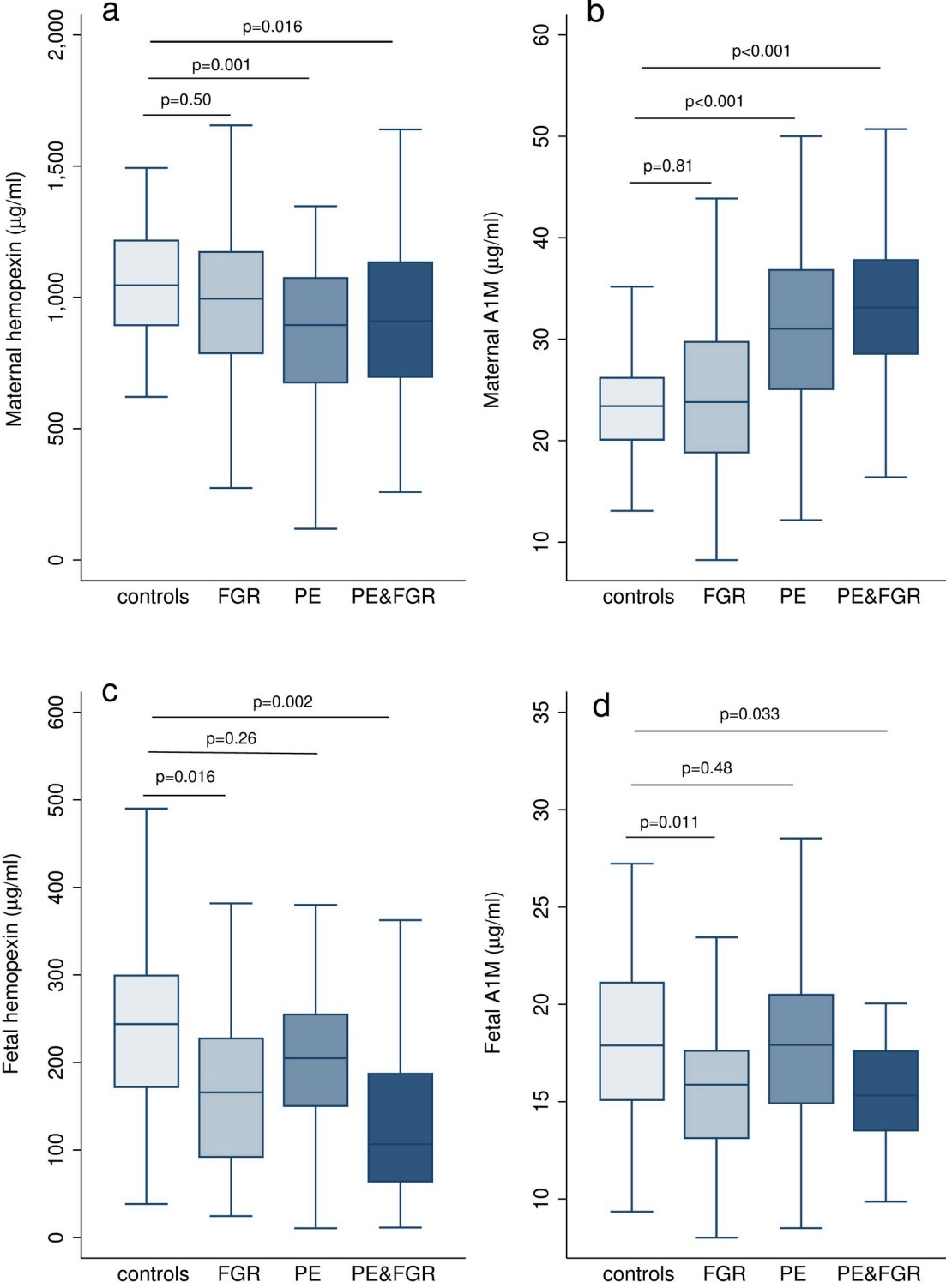

**Fig 2. Concentrations of heme scavengers in maternal and fetal blood in the study population.** Box plots represent a) maternal hemopexin concentrations, b) maternal A1M concentrations, c) fetal hemopexin concentrations, d) fetal A1M concentrations. Boxes show median and interquartile range, whiskers represent 1.5 X interquartile range or the extremes of the distribution. FGR, fetal growth restriction; PE, preeclampsia; A1M, $\alpha_1$-microglobulin. p values were calculated by multiple regression (multivariate linear regression) compared to controls, adjusted for chronic hypertension, diabetes, assisted reproductive technologies, smoking, fetal sex and gestational age at sampling.

the fetus in contrast with depletion of hemopexin in both. A possible explanation to the difference in the behavior of hemopexin and A1M may be their different clearance routes. While hemopexin-heme complexes get cleared through a specific receptor [33], A1M clearance is dependent on the kidney filtration [34]. The filtration capacity becomes reduced in PE mothers while in the developing fetus the placenta takes over the role of the kidney and permitting A1M clearance [35, 36]. No correlation existed between maternal and fetal heme scavengers, indicating that the concentrations of these biomarkers in maternal and fetal circulations are independent of each other and reflect the status of the mother and the fetus, separately. Thus, the heme scavenger systems seem to be affected in the mother or the fetus in isolated PE or FGR, respectively, and in both the mother and the fetus when the two disorders coexist. Free HbF, LDH activity and erythropoietin results were concordant reflecting a status of chronic hypoxia in FGR fetuses particularly those from PE pregnancies.

## Results of the study in the context of other observations

In recent publications, PE has been associated with increased levels of cell-free hemoglobin and A1M together with lower haptoglobin and hemopexin [14, 21]. Our data are consistent with these reports, and provide further evidence that both PE with and without FGR have the same profile in maternal blood. In addition to this, we now show that there are no differences in hemopexin and A1M levels between normotensive FGR mothers and controls. In line with our results, studies have shown that the heme scavengers profile was similar in early and late-onset PE, and have thus suggested them as biomarkers for the disorder [8, 21]. Their role in predicting PE has also been investigated, showing that higher A1M and lower hemopexin levels were present already from the first trimester in pregnancies that later developed PE [19, 20]. Moreover, a recent study showed that high-risk pregnancies without PE also had an altered profile of heme scavengers compared to low-risk gestations [31]. Free HbF and heme scavenger concentrations in cord blood were first explored by Brook et al showing an elevated level of free HbF in FGR fetuses, which may contribute to increased fetoplacental vascular resistance and impaired endothelial protection [9]. In the current study, we further demonstrate that FGR fetuses have a similar profile of heme scavengers regardless of whether the mother is normotensive or preeclamptic. Additionally, we show that fetuses from PE pregnancies without FGR did not present any difference compared to controls. These findings were independent of fetal sex in line with previous reports on comparable free HbF levels among male and female fetuses [37, 38].

## Pathophysiological mechanisms

Several studies have suggested that PE and FGR may originate in the first trimester developing placental insufficiency associated with local hypoxia [5, 6]. Hypoxia, in turn, may lead to increased erythropoiesis through the increased synthesis of erythropoietin, and induce a switch from adult hemoglobin to HbF in cord blood cells [39, 40]. Free hemoglobin might participate in oxidative damage to placental tissues and contribute to FGR and fetal compromise [9]. In normotensive FGR, the damage may remain local, affecting primarily the developing fetus, whereas in cases of disrupted blood–placenta barrier, free hemoglobin could leak to the maternal circulation exerting systemic oxidative stress on the endothelium. Thus, free hemoglobin in the maternal circulation could be fetal hemoglobin originating from the placenta, or adult free hemoglobin originating from hemolysis of maternal red blood cells [14, 41]. In both cases, free hemoglobin contributes to the oxidative stress and vasoconstriction seen in PE. In the other phenotype of PE, PE without FGR, the heme scavenger system's alterations could be explained by maternal systemic maladaptation to the pregnancy manifesting by the maternal

disease with subsequent late placental involvement [5, 42, 43]. In this case, the blood-placenta barrier remains intact and the fetus stays unaffected with a clinical presentation as late-onset PE principally.

## Strengths and limitations

This was a prospective study with meticulous characterization of the included pregnancies in addition to the collection of paired maternal and fetal blood samples at the time of delivery. However, it was unfeasible to match cases with controls for gestational age at time of sampling due to the greater proportion of elective deliveries in PE and/or FGR. We believe that the perceived differences in heme scavengers could not be explicated by the difference in gestational age at delivery, since the concentrations of hemopexin and A1M have been described to be altered from the first trimester in pregnancies that developed PE [19, 20]. Moreover, we applied careful statistical analysis adjusting our findings for potential confounders comprising the gestational age. The current study was substantially composed of late-onset pregnancies, which constitute the most prevalent form of placenta-mediated disorders, with quite a few early-onset cases. Although we focused on hemopexin and A1M in this study, we believe that assessing the albumin, other heme scavengers such as the haptoglobin and markers of oxidative stress like oxidized lipids would be of interest. Moreover, investigating the changes in heme scavengers through the pregnancy would complement our knowledge in terms of their role in the early stages of gestation.

## Conclusions and perspectives

Our results support the hypothesis that hemoglobin-induced oxidative stress contribute to the pathogenesis in both PE and FGR. In addition, we highlight the differential involvement of heme scavenger systems, which are responsible for protecting against this oxidative stress, in accordance with the clinical presentation of PE and/or FGR. The scavengers hemopexin and A1M presented significantly altered concentrations in maternal blood in the maternal disease -PE- and in cord blood in the fetal disease -FGR-. Further research is warranted to identify the precise effects of hemoglobin-induced oxidative stress on the mother and the fetus, to clarify the placental behavior in the different phenotypes of PE and/or FGR and to determine potential interventional measures for the treatment or prevention of these disorders and their related complications whereby scavenger protein-based therapeutics are potential candidates [12, 44]. Of interest, recombinant A1M has shown a protective role against organ damage and FGR in preeclampsia mouse model, ewe and rabbits as well [45–47]. Human studies are necessitated to investigate its applicability on PE patients and its potential role in FGR [44, 48].

## Supporting information

**S1 File.**
(PDF)

## Acknowledgments

This research used the Hospital Clínic-IDIBAPS Biobank resource and the Clinical Sciences Department facilities at Lund University. We thank the patients for participating in this study and the nurses from the Departments of Maternal-Fetal Medicine at BCNatal (Barcelona, Spain) for their help in collecting human samples. We also thank Eva Hansson and Malgorzata Berlikowski from the Department of Clinical Sciences, Lund University, for their contribution in the lab work.

## Author Contributions

**Conceptualization:** Lina Youssef, Fatima Crispi, Stefan R. Hansson.

**Data curation:** Lina Youssef.

**Formal analysis:** Lina Youssef.

**Funding acquisition:** Bo Åkerström, Fatima Crispi, Eduard Gratacos.

**Investigation:** Lina Youssef, Jezid Miranda, Cristina Paules, Francesca Crovetto.

**Methodology:** Lina Youssef, Lena Erlandsson, Bo Åkerström, Stefan R. Hansson.

**Project administration:** Fatima Crispi, Stefan R. Hansson.

**Resources:** Bo Åkerström, Stefan R. Hansson.

**Supervision:** Lena Erlandsson, Fatima Crispi, Eduard Gratacos, Stefan R. Hansson.

**Validation:** Lina Youssef, Lena Erlandsson, Stefan R. Hansson.

**Visualization:** Lina Youssef.

**Writing – original draft:** Lina Youssef.

**Writing – review & editing:** Lina Youssef, Lena Erlandsson, Bo Åkerström, Fatima Crispi, Eduard Gratacos, Stefan R. Hansson.

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
