## [Decision Letter · Decision Letter 0]

19 Jun 2020

PONE-D-20-14717

Hemopexin and α1-microglobulin heme scavengers with differential involvement in preeclampsia and fetal growth restriction

PLOS ONE

Dear Dr. Youssef,

Thank you for submitting your manuscript to PLOS ONE. After careful consideration, we feel that it has merit but does not fully meet PLOS ONE’s publication criteria as it currently stands. Therefore, we invite you to submit a revised version of the manuscript that addresses the points raised during the review process.

SPECIFIC ACADEMIC EDITOR COMMENTS: There were two expert reviewers in the field that handled your manuscript. We thank them for their time. Although interest was found in your study, several major points arose during review that require your attention. Comments that need addressing include: 1) several questions about the methods, 2) the need for correlative studies, and 3) measures of oxidative stress. Please address all of the reviewers' comments in your revised manuscript.

We look forward to receiving your revised manuscript.

Kind regards,

Frank T. Spradley

Academic Editor

PLOS ONE

2. We note that you have a patent relating to material pertinent to this article. Please provide an amended statement of Competing Interests to declare this patent (with details including name and number), along with any other relevant declarations relating to employment, consultancy, patents, products in development or modified products etc. Please confirm that this does not alter your adherence to all PLOS ONE policies on sharing data and materials, as detailed online in our guide for authors http://journals.plos.org/plosone/s/competing-interests by including the following statement: "This does not alter our adherence to  PLOS ONE policies on sharing data and materials.” If there are restrictions on sharing of data and/or materials, please state these. Please note that we cannot proceed with consideration of your article until this information has been declared.

3. Thank you for submitting the above manuscript to PLOS ONE. During our internal evaluation of the manuscript, we found some minor text overlap between your Discussion section (limitations) of your submission and the following previously published work:

https://www.ajog.org/article/S0002-9378(19)30909-3/fulltext

Please revise the manuscript to rephrase the duplicated text and ensure you cite your sources.

4. Please report the source, catalog number, and dilutions of all antibodies used in your study. Please also report the catalog number of all commercial kits and immunoassays used.

Reviewers' comments:

Reviewer's Responses to Questions

**Comments to the Author**

1. Is the manuscript technically sound, and do the data support the conclusions?

Reviewer #1: Yes

Reviewer #2: Yes

2. Has the statistical analysis been performed appropriately and rigorously? 

Reviewer #1: Yes

Reviewer #2: Yes

3. Have the authors made all data underlying the findings in their manuscript fully available?

Reviewer #1: Yes

Reviewer #2: Yes

4. Is the manuscript presented in an intelligible fashion and written in standard English?

Reviewer #1: Yes

Reviewer #2: Yes

5. Review Comments to the Author

Reviewer #1: The study presented by Youssef et al. evaluated maternal and fetal plasma

concentrations of heme scavengers (Hemopexin and α1-microglobulin) in different phenotypes of placenta-mediated disorders and controls, both in mothers and neonates.

To prevent the toxic, and notably, the pro-oxidant effect of free hemoglobin and its degradation compounds haem and free iron, the human body disposes of circulating scavenger proteins, including notably those two molecules, that have been associated with preeclampsia.

The study includes 47 singleton pregnancies collected in a monocentric prospective study.

Its originality is related to the exploration of those two haem scavengers both mothers and neonates groups in a same study with the same methods, compared to previously published study.

The manuscript is overall well written and clear. The population is well described, the mention of potential treatments for PE should be cited.

The limits of the paper are clearly exposed.

The most surprising result is the difference observed for A1 microglobulin between mothers and fetuses, in contrast with hemopexin. This point is debated in the discussion section.

Main corrections

A flow chart of the study cohort should be presented.

The quality of the figure and legend should be increased. The legend has to permit the reader to understand them without the text. What type of graph is presented? Median IQ? A line showing which populations have been compared with p should be easier to read.

The analysis of the quality of the homemade A1M assay should be presented briefly. Notably, the standard curve should be presented in supporting pieces of information, or a previous publication should be cited, if it do exist. The manufacturer’s providing the monoclonal antibodies should be mentioned. Whether all samples were analyzed in a single batch and if some internal quality controls were included should be precised (since unknown samples are cited).

As the renal function is supposed to explain the increase of A1M in mothers, an analysis of A1M concentration regarding the creatinine concentration would possibly reinforce this hypothesis. Some additional explanations should be explored, such as albumin and haptoglobin concentrations in maternal and fetal bloods.

Minor corrections

The population is extremely well described. However, some data could be suppressed because there are discussed further. For instance, the classification between mild and severe is not use in the analysis of the data, so I’m not sure it deserves to be mentioned in this setting.

It should be mentioned whether the assays were done in singlicates for hemopexin.

Units have to be corrected ug/mL replace by �g/mL using symbol font.

The legend of the supporting tables has to be clarified: I think only mean + Sd is presented for biological values, whereas we can read “Data are mean (Å} standard deviation) or median (interquartile range).”

Line 131 : three instead of 3

Line 132 : two instead of 2

Reviewer #2: Overall, well performed set of experiments. I have only minor comments.

1. There's obviously a wide range of lab values in each group. Was any attempt made to correlate the levels of A1M or hemopexin with overall outcomes? Time to delivery, maternal morbidity, et cetera.

2. Was any attempt made to measure markers of oxidative stress? Oxidized lipids, for instance?

6. PLOS authors have the option to publish the peer review history of their article (what does this mean?). If published, this will include your full peer review and any attached files.

Reviewer #1: No

Reviewer #2: No

---

## [Author Response · Author response to Decision Letter 0]

9 Jul 2020

Dear Editor,

Thank you very much for considering our manuscript “Hemopexin and α1-microglobulin heme scavengers with differential involvement in preeclampsia and fetal growth restriction” (PONE-D-20-14717).

We would like to thank the reviewers for their detailed comments and suggestions for the manuscript. We believe that the comments have identified important areas which required improvement. Below, you will find a point by point description of how each comment was addressed in the manuscript.

ACADEMIC EDITOR COMMENTS: 

There were two expert reviewers in the field that handled your manuscript. We thank them for their time. Although interest was found in your study, several major points arose during review that require your attention. Comments that need addressing include: 1) several questions about the methods, 2) the need for correlative studies, and 3) measures of oxidative stress. Please address all of the reviewers' comments in your revised manuscript.

Answer: 1) the methods’ details requested by the reviewers have been added as specified below, 2) correlative studies have been added to the manuscript, 3) unfortunately, other biomarkers of oxidative stress have not been assessed in this study (this comment has been added to the limitations section).

Answer: the style and file names have been modified according to these instructions.

2. We note that you have a patent relating to material pertinent to this article. Please provide an amended statement of Competing Interests to declare this patent (with details including name and number), along with any other relevant declarations relating to employment, consultancy, patents, products in development or modified products etc. Please confirm that this does not alter your adherence to all PLOS ONE policies on sharing data and materials, as detailed online in our guide for authors http://journals.plos.org/plosone/s/competing-interests by including the following statement: "This does not alter our adherence to PLOS ONE policies on sharing data and materials.” If there are restrictions on sharing of data and/or materials, please state these. Please note that we cannot proceed with consideration of your article until this information has been declared.

Answer: We added the patent details as follows. “SH and BÅ hold a patent related to the subject in this paper (PCT appl no. WO2008098734A1; Diagnosis and treatment of preeclampsia). SH and BÅ are co-founders and share-holders of Guard Therapeutics formerly named A1M Pharma. This company develops a treatment of acute kidney injury based on α1-microglobulin, but is not involved in clinical development of diagnosis and treatment of preeclampsia, and has not supported this study financially. This does not alter our adherence to PLOS ONE policies on sharing data and materials. The remaining authors declare that the research was conducted in the absence of any commercial or financial relationships that could be construed as a potential conflict of interest.”

3. Thank you for submitting the above manuscript to PLOS ONE. During our internal evaluation of the manuscript, we found some minor text overlap between your Discussion section (limitations) of your submission and the following previously published work:

https://www.ajog.org/article/S0002-9378(19)30909-3/fulltext

Please revise the manuscript to rephrase the duplicated text and ensure you cite your sources.

Answer: the strengths and limitations section has been modified according to these instructions (lines 430-498).

4. Please report the source, catalog number, and dilutions of all antibodies used in your study. Please also report the catalog number of all commercial kits and immunoassays used.

Answer: These information have been added in the methods section (lines 205-259). 

Answer: Thank you for providing this resource. This check has been made and figures modified by PACE have been submitted.

Reviewers' comments: 

REVIEWER 1

The study presented by Youssef et al. evaluated maternal and fetal plasma concentrations of heme scavengers (Hemopexin and α1-microglobulin) in different phenotypes of placenta-mediated disorders and controls, both in mothers and neonates.

To prevent the toxic, and notably, the pro-oxidant effect of free hemoglobin and its degradation compounds haem and free iron, the human body disposes of circulating scavenger proteins, including notably those two molecules, that have been associated with preeclampsia.

The study includes 47 singleton pregnancies collected in a monocentric prospective study.

Its originality is related to the exploration of those two haem scavengers both mothers and neonates groups in a same study with the same methods, compared to previously published study.

The manuscript is overall well written and clear. The population is well described, the mention of potential treatments for PE should be cited.

Answer: The corresponding citation has been added (line 512) 

The limits of the paper are clearly exposed.

The most surprising result is the difference observed for A1 microglobulin between mothers and fetuses, in contrast with hemopexin. This point is debated in the discussion section.

Main corrections

1. A flow chart of the study cohort should be presented.

Answer: The flow chart of the study has been added to the manuscript as Fig1.

2. The quality of the figure and legend should be increased. The legend has to permit the reader to understand them without the text. What type of graph is presented? Median IQ? A line showing which populations have been compared with p should be easier to read.

Answer: The image quality has been increased (currently it’s provided in .eps format). The type of graph has been added to the legend indicating that it is a box plot that represent median and IQ range. A line showing p values has been included in the figure. In addition, the type of graph has also been added to the legend of S3 Fig.

3. The analysis of the quality of the homemade A1M assay should be presented briefly. Notably, the standard curve should be presented in supporting pieces of information, or a previous publication should be cited, if it do exist. The manufacturer’s providing the monoclonal antibodies should be mentioned. Whether all samples were analyzed in a single batch and if some internal quality controls were included should be precised (since unknown samples are cited).

Answer: Maternal and fetal A1M average intraassay coefficient of variation were 2.23% (�1.8% standard deviation) and 1.21% (�0.95% standard deviation) respectively. The samples were analyzed in 10 batches (5 maternal and 5 fetal). A new standard curve has been prepared for each plate, an example has been added as S1 Fig with a comment in lines 352-357. The interassay coefficient of variance was assessed in a previous study by Anderson et al. Am J Obstet Gynecol 2011;204:520.e1-5 (it was <4.05%). In the current study, the interassay coefficient of variation has not been assessed. The manufacturer’s providing the monoclonal antibodies has been added in line 358.

4. As the renal function is supposed to explain the increase of A1M in mothers, an analysis of A1M concentration regarding the creatinine concentration would possibly reinforce this hypothesis. Some additional explanations should be explored, such as albumin and haptoglobin concentrations in maternal and fetal bloods.

Answer: Thank you for the relevant comment. In fact, we have data concerning creatinine levels in PE mothers since it is measured routinely for clinical purposes, but no data are available for controls, normotensive FGR mothers neither for fetuses from all the study groups. A1M was correlated with creatinine in PE mothers in contrast to no correlation between hemopexin and creatinine. This correlation analysis has been added to the manuscript as S2 Fig. Most unfortunately, Albumin and haptoglobin concentrations were not assessed in the study population. We added this comment to the limitations section line 495.

Minor corrections

1. The population is extremely well described. However, some data could be suppressed because there are discussed further. For instance, the classification between mild and severe is not use in the analysis of the data, so I’m not sure it deserves to be mentioned in this setting.

Answer: We agree with the reviewer that some data might be suppressed. The classification to mild and severe preeclampsia has been described in the manuscript to relate it with the sub analysis on preterm and term pregnancies given that the vast majority of preterm cases are severe preeclampsia. However, the criteria of severe preeclampsia have been suppressed with the appropriate reference (line 174).

2. It should be mentioned whether the assays were done in singlicates for hemopexin.

Answer: The hemopexin assays were done in duplicates. A comment has been added in the methods section (line 207). 

3. Units have to be corrected ug/mL replace by �g/mL using symbol font.

Answer: The units have been corrected in both Fig 2 and S3 Fig. 

4. The legend of the supporting tables has to be clarified: I think only mean + Sd is presented for biological values, whereas we can read “Data are mean (Å} standard deviation) or median (interquartile range).”

Answer: The legend of the supporting tables has been clarified, and “Data are mean (Å} standard deviation) or median (interquartile range)” has been modified into “Data are mean (Å} standard deviation)”.

5. Line 131 : three instead of 3; Line 132 : two instead of 2.

Answer: These changes have been made (lines 171-172 in the new version with track changes). 

REVIEWER 2

Overall, well performed set of experiments. I have only minor comments.

1. There's obviously a wide range of lab values in each group. Was any attempt made to correlate the levels of A1M or hemopexin with overall outcomes? Time to delivery, maternal morbidity, et cetera.

Answer: Yes, we have explored the correlation/association between hemopexin or A1M and the pregnancy outcome. None of these two heme scavengers were correlated with time to delivery, maternal morbidity, including severity and days of admission at the intensive care unit, or perinatal morbidity such as low APGAR score at 5 minutes and neonatal intensive care unit admissions. This comment has been added to the manuscript (lines 340-342). 

2. Was any attempt made to measure markers of oxidative stress? Oxidized lipids, for instance?

Answer: Unfortunately, these markers have not been assessed in the present study. This comment has been added to the limitations section line 496.

Thank you for your consideration!

Sincerely,

Lina Youssef, MD PhD

Erasmus Mundus Doctorate in Fetal and Perinatal Medicine – FetalMed-PhD

---

## [Decision Letter · Decision Letter 1]

25 Aug 2020

PONE-D-20-14717R1

Hemopexin and α1-microglobulin heme scavengers with differential involvement in preeclampsia and fetal growth restriction

PLOS ONE

Dear Dr. Youssef,

Thank you for submitting your manuscript to PLOS ONE. After careful consideration, we feel that it has merit but does not fully meet PLOS ONE’s publication criteria as it currently stands. Therefore, we invite you to submit a revised version of the manuscript that addresses the points raised during the review process.

ACADEMIC EDITOR COMMENTS: There are some remaining comments that must be addressed by the authors in your revised manuscript.

We look forward to receiving your revised manuscript.

Kind regards,

Frank T. Spradley

Academic Editor

PLOS ONE

Reviewers' comments:

Reviewer's Responses to Questions

**Comments to the Author**

1. If the authors have adequately addressed your comments raised in a previous round of review and you feel that this manuscript is now acceptable for publication, you may indicate that here to bypass the “Comments to the Author” section, enter your conflict of interest statement in the “Confidential to Editor” section, and submit your "Accept" recommendation.

Reviewer #1: All comments have been addressed

2. Is the manuscript technically sound, and do the data support the conclusions?

Reviewer #1: Yes

3. Has the statistical analysis been performed appropriately and rigorously? 

Reviewer #1: Yes

4. Have the authors made all data underlying the findings in their manuscript fully available?

Reviewer #1: Yes

5. Is the manuscript presented in an intelligible fashion and written in standard English?

Reviewer #1: Yes

6. Review Comments to the Author

Reviewer #1: Thank you for this revised version, that is now for me perfectly acceptable for publication.

I just notice the following mistake to correct :

Legend Fig 1 ; FGR, fetal grown restriction : replace by fetal growth restriction

7. PLOS authors have the option to publish the peer review history of their article (what does this mean?). If published, this will include your full peer review and any attached files.

Reviewer #1: **Yes: **Katell Peoc'h

---

## [Author Response · Author response to Decision Letter 1]

27 Aug 2020

ACADEMIC EDITOR COMMENTS: 

There are some remaining comments that must be addressed by the authors in your revised manuscript. 

Answer: The reviewer comments have been addressed and the mistake has been corrected from “grown” to “growth”.

Reviewers' comments: 

REVIEWER 1

Thank you for this revised version, that is now for me perfectly acceptable for publication.

I just notice the following mistake to correct : Legend Fig 1 ; FGR, fetal grown restriction : replace by fetal growth restriction

Answer: The mistake has been corrected in Fig 1 legend as well as Table 1 footnote and other tables in the supporting information.

---

## [Editor Report · Decision Letter 2]

31 Aug 2020

Hemopexin and α1-microglobulin heme scavengers with differential involvement in preeclampsia and fetal growth restriction

PONE-D-20-14717R2

Dear Dr. Youssef,

We’re pleased to inform you that your manuscript has been judged scientifically suitable for publication and will be formally accepted for publication once it meets all outstanding technical requirements.

Kind regards,

Frank T. Spradley

Academic Editor

PLOS ONE

---

## [Editor Report · Acceptance letter]

2 Sep 2020

PONE-D-20-14717R2 

Hemopexin and α^1^-microglobulin heme scavengers with differential involvement in preeclampsia and fetal growth restriction 

Dear Dr. Youssef:

I'm pleased to inform you that your manuscript has been deemed suitable for publication in PLOS ONE. Congratulations! Your manuscript is now with our production department. 

Kind regards, 

on behalf of

Dr. Frank T. Spradley 

Academic Editor

PLOS ONE